# `BotPercent`: Estimating Bot Populations in Twitter Communities

**Zhaoxuan Tan**[*♣]    **Shangbin Feng**[*♠]    **Melanie Sclar**[♠]    **Herun Wan**[♡]
**Minnan Luo**[♡]    **Yejin Choi**[♠♢]    **Yulia Tsvetkov**[♠]
University of Notre Dame[♣], University of Washington[♠]
Xi'an Jiaotong University[♡], Allen Institute for AI[♢]
ztan3@nd.edu; shangbin@cs.washington.edu

## Abstract

Twitter bot detection is vital in combating mis-information and safeguarding the integrity of social media discourse. While malicious bots are becoming more and more sophisticated and personalized, standard bot detection approaches are still agnostic to social environments (henceforth, *communities*) the bots operate at. In this work, we introduce **community-specific bot detection**, estimating the percentage of bots given the context of a community. Our method—`BotPercent`—is an amalgamation of Twitter bot detection datasets and feature-, text-, and graph-based models, adjusted to a particular community on Twitter. We introduce an approach that performs confidence calibration across bot detection models, which addresses generalization issues in existing community-agnostic models targeting individual bots and leads to more accurate community-level bot estimations. Experiments demonstrate that `BotPercent` achieves state-of-the-art performance in community-level Twitter bot detection across both balanced and imbalanced class distribution settings, presenting a less biased estimator of Twitter bot populations within the communities we analyze. We then analyze bot rates in several Twitter groups, including users who engage with partisan news media, political communities in different countries, and more. Our results reveal that the presence of Twitter bots is not homogeneous, but exhibiting a spatial-temporal distribution with considerable heterogeneity that should be taken into account for content moderation and social media policy making. The implementation of `BotPercent` is available at https://github.com/TamSiuhin/BotPercent.

## 1 Introduction

Twitter accounts controlled by automated programs, also known as Twitter *bots*, have become a widely recognized, concerning, and studied phenomenon (Ferrara et al., 2016a; Aiello et al., 2012).

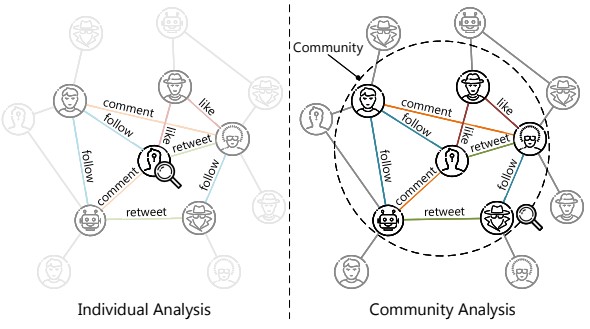

Figure 1: Beyond individual bot detection, our work proposes a socially contextualized community-level bot detection and analysis of bot populations within different communities on Twitter.

Twitter bots have been deployed with malicious intents, such as disinformation spread (Cui et al., 2020; Wang et al., 2020; Lu and Li, 2020; Huang et al., 2022), interference in elections (Howard et al., 2016; Bradshaw et al., 2017; Ferrara et al., 2016a; Rossi et al., 2020), promotion of extremism (Ferrara et al., 2016b; Marcellino et al., 2020), and the spread of conspiracy theories (Ferrara, 2020; Ahmed et al., 2020). These triggered the development of automatic Twitter bot detection models aiming at mitigating harms from bots' malicious interference (Cresci, 2020).

Prior work has mainly focused on bot detection at the individual account level (Yang et al., 2020; Echeverrıa et al., 2018; Feng et al., 2022a), whereas community-level estimates of bot population and activity are under-explored, where "community" is defined as network proximity in line with prior works (Feng et al., 2021b, 2022b). Our work posits that the social context in which a bot operates is essential for accurate detection, with major implications for both the social media platform and users. From the platform side, it can allow decision-makers to efficiently distribute content moderation resources among communities, as well as inform community members about the risks of inauthen-

---

*These authors contributed equally to this work.

tic content. From a user perspective, it can enhance awareness of potential opinion manipulation by clearly indicating the anticipated level of inauthentic content within a social network. Finally, community-level bot detection can help alleviate privacy concerns by presenting collective statistics per-community, rather than probing or tracking individual users within the community, as required by existing approaches to individual bot detection (Guerid et al., 2013). These and other commercial and legal concerns have led to an increased interest in understanding the percentage of Twitter bots from groups to crowds (Varol, 2022).

Although state-of-the-art Twitter bot detection methods have achieved impressive results, their individual-bot focus renders them unsuitable for out-of-the-box population-level predictions. Current approaches often employ models that analyze a single modality of user information in a single dataset, overfitting to certain types of Twitter bots (Echeverrıa et al., 2018; Yang et al., 2020). Such individual-bot detection approaches are often poorly calibrated and thus cannot be easily adapted to estimating bot populations, since robust community-level bot detection requires having an unbiased estimator of bot probability.

To this end, we propose BotPercent, a framework for community-oriented bot detection that aggregates and calibrates multiple existing models and datasets to devise accurate and generalizable estimations of Twitter bot populations from groups to crowds. BotPercent combines feature, text, and graph-based approaches with complementary inductive biases, which facilitates robustness to shifting user domains. This ensemble aims to balance the over-specialization of single-modality models. Since our analysis shows that community-agnostic models are often miscalibrated, BotPercent also conducts model calibration for individual models (Guo et al., 2017) and combines their predictions while dynamically learning the weights of individual models in shifting contexts for more accurate bot percentage estimation. For training data, BotPercent merges multiple available Twitter bot detection datasets with 1,216,758 users to enhance generalization and overcome the data limitations of existing community-agnostic approaches.

We first evaluate BotPercent with the 10 Twitter community datasets of the TwiBot-22 benchmark (Feng et al., 2022b) through both balanced and imbalanced class distribution community settings. Extensive experiments demonstrate that BotPercent achieves state-of-the-art performance for community-level bot detection, presenting consistently more accurate and calibrated bot population estimations across both settings in ten diverse Twitter communities. Armed with BotPercent, we investigate the bot populations in real-world Twitter communities of varying sizes and contexts, including political groups, news-sharing communities, celebrity-centric cliques, and more. We find that (1) around 8-14% of interactions with Elon Musk's Twitter poll[1] to reinstate Donald Trump were carried out by bots; (2) online communities that focus on politics and cryptocurrency are witnessing an elevated level of bot interference; (3) news media that are more partisan often face higher rates of bot-generated comments; (4) bot populations in the democratic discourse around political issues (e.g., abortion and immigration) often wax and wane due to major socio-political events. Together our experiments and results demonstrate that the distribution of Twitter bots is not homogeneous, but rather has spatial-temporal patterns with significant implications for bot behavior understanding, navigating socio-political events, social media moderation, and more.

The contribution of BotPercent is the largest and most comprehensive combination of existing bot detection models and datasets with confidence calibration, addressing a novel problem of community-level bot detection which provides better aggregated estimates of bot populations within social networks, leading to new findings while avoiding the privacy risks of tracking individual accounts.

## 2 BotPercent Methodology

We propose **BotPercent**, a novel system for community-level Twitter bot detection. As illustrated in Figure 2, BotPercent first trains multiple models on a mixture of datasets, then leverages confidence calibration to perform unbiased bot percentage estimation, and finally combines predictions with learnable weights.

### 2.1 Model Components

Motivated by the fact that different types of bot detection models have their strengths and weaknesses in the face of customized bots in different

---

[1] https://twitter.com/elonmusk/status/1593767953706921985

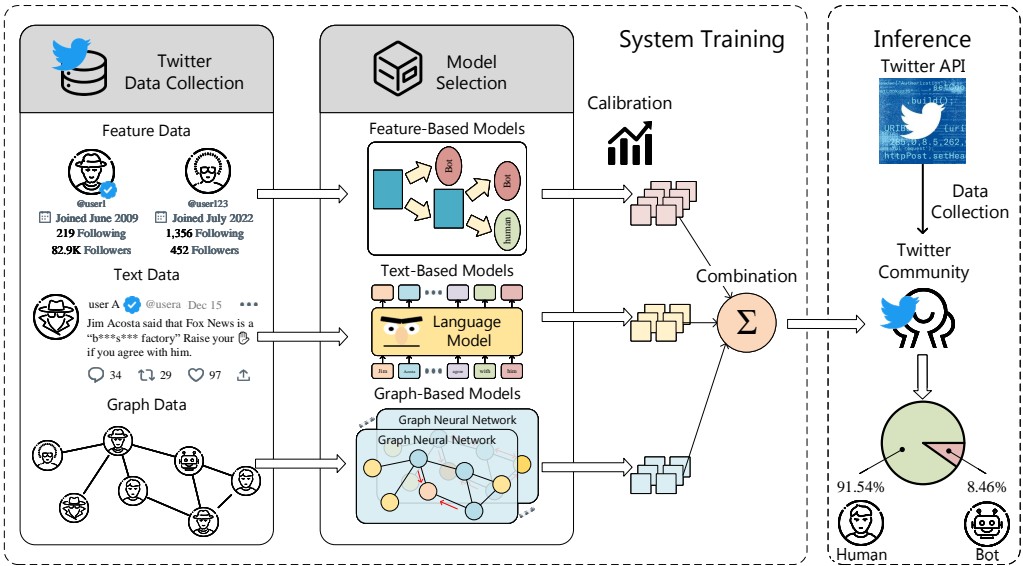

Figure 2: Overview of `BotPercent`. `BotPercent` first trains multiple models on merged datasets and then uses confidence calibration to conduct accurate bot percentage estimation. Finally, it combines the predictions with learnable weights to obtain the final result.

communities (Sayyadiharikandeh et al., 2020), we propose a unified framework that adopts representative approaches. Existing Twitter bot detection approaches could be categorized based on their input data: feature-, text-, and graph-based models use different kinds of user features, adopt different classifier architectures, and have respective inductive biases (Feng et al., 2022b). We combine and calibrate these as modular components of `BotPercent`, enhancing individual models' performance and generalizability.

**Feature-based** models extract user features and adopt traditional classifiers (Varol et al., 2017). To construct a comprehensive feature-based model as part of `BotPercent`, we summarize features introduced in existing feature-based models and obtain a more comprehensive feature set consisting of 12 direct and 14 derived user features. Following previous works (Yang et al., 2020; Knauth, 2019), `BotPercent` leverages random forest (Ho, 1995) and AdaBoost (Freund and Schapire, 1997) as two efficient feature-based classifiers and obtains binary prediction logits $p_f \in \mathbb{R}^{1 \times 2}$ for each Twitter user representing its probabilities as bot or human.

**Text-based** bot detection models leverage users' tweets and descriptions to identify Twitter bots and malicious content (Wei and Nguyen, 2019). Similarly, `BotPercent` leverages pretrained RoBERTa (Liu et al., 2019a) and T5 (Raffel et al., 2020) language models to encode user descriptions and 20 latest tweets while using a linear layer Linear for

classification:

$$p_t = \text{Linear}([\text{LM}(\{\mathbf{t}_i\}_{i=1}^{L_t})||\text{LM}(\{\mathbf{d}_i\}_{i=1}^{L_d})]),$$

where $p_t \in \mathbb{R}^{1 \times 2}$ denotes the prediction logits, $[\cdot||\cdot]$ denotes vector concatenation, $\mathbf{t}_i$ and $\mathbf{d}_i$ represent tweet and user description tokens respectively, and LM denotes one of the language models with a mean pooling feature extractor over the final layer. These text-based models are optimized with the cross entropy loss function.

**Graph-based** bot detection models leverage the Twitter network structure with graph neural networks (GNNs) to analyze the contextual user interactions through local neighborhood information aggregation (Ali Alhosseini et al., 2019; Feng et al., 2022a). For graph-based models, we employ four graph neural network-based approaches to bot detection: SimpleHGN (Lv et al., 2021), HGT (Hu et al., 2020), BotRGCN (Feng et al., 2021c), and RGT (Feng et al., 2022a) in `BotPercent` since these models take into account the intrinsic heterogeneity in social networks and have shown promising bot detection performance (Feng et al., 2022b). The message-passing paradigm of these models could be summarized as:

$$\mathbf{h}_i^{(l)} = \underset{\forall j \in \mathcal{N}_i, \forall e \in \mathcal{E}(i,j)}{\text{Agg.}^{(l)}} (\text{Extr.}^{(l)}(\mathbf{h}_i^{(l-1)}, \mathbf{h}_j^{(l-1)}, \mathbf{e})),$$

where $\mathbf{h}_i^{(l)}$ denotes the $i$-th user's representation in the $l$-th GNN layer, $\mathcal{N}_i$ represents the neighbor nodes of user $i$, and $\mathcal{E}(i, j)$ denotes all the edges

from user $j$ to $i$. Extr.$^{(l)}$ represents the neighbor information extractor in $i$-th layer, which extracts user information from source user's representation $\mathbf{h}_j^{(l-1)}$, with the target user representation $\mathbf{h}_i^{(l-1)}$ and the edge $e$ between two users as propagated message. Agg.$^{(l)}$ gathers the neighborhood information of source users via aggregation operators. Different GNNs use different aggregation and extraction functions. SimpleHGN uses the attention mechanism with consideration of edge type as Agg.$^{(l)}$ and an MLP as Extr.$^{(l)}$. HGT adopts the attention mechanism with regard to edge type as Agg.$^{(l)}$ and takes the edge type as different projection matrixes in Extr.$^{(l)}$. BotRGCN takes the mean pooling as Agg.$^{(l)}$ and processes the edge type with different aggregation matrixes in Extr.$^{(l)}$. RGT propagates message under different relation types as Extr.$^{(l)}$ and aggregate representation from different relation types with the attention mechanism in Agg.$^{(l)}$. With these different GNN architectures and message-passing mechanisms, we aim to capture the diverse interaction patterns between multi-faceted bots and users.

After modeling the Twitter network with $L$ layers of GNNs, we obtain the representation $\mathbf{h}_i^{(L)}$ for user $i$. BotPercent then employs a linear layer for two-way classification as $p_g = \text{Linear}(\mathbf{h}_i^{(L)})$. Graph-based approaches are optimized with cross entropy loss function.

A unique challenge of employing graph-based approaches for community-level bot detection is that they face scalability issues at inference time due to data dependency: when BotPercent analyzes a specific user, it needs to encode its multihop following/follower neighborhood, which leads to exponential inference costs. Motivated by Zhang et al. (2021), we use knowledge distillation (Hinton et al., 2015) to transfer the knowledge of graph-based detectors to efficient linear layers. The distillation training loss could be written as:

$$L_d = \lambda \sum_{v \in \mathcal{V}} \text{CE}(\hat{y}_v, y_v) + (1 - \lambda) \sum_{v \in \mathcal{V}} \text{KL}(\hat{y}_v || p_g^v),$$

where $\mathcal{V}$ denotes a batch of users, CE denotes the cross entropy loss, KL denotes the KL-divergence, $\lambda$ is a hyperparameter, $\hat{y}_v$, $p_g^v$, and $y_v$ denotes the prediction of the linear layer, GNNs, and the ground truth label. In this way, state-of-the-art graph-based approaches are distilled into high-quality linear layers that serve as proxies for the computation-heavy GNNs, improving

BotPercent's efficiency and scalability for the large-scale real-time analysis of social network communities.

## 2.2 Confidence Calibration

Existing bot detection models generally leverage one dataset as training data, and existing datasets are limited in user domains, Twitter communities, and data collection times (Feng et al., 2021b). Thus, current models generalize poorly to new user communities and emerging bots (Echeverría et al., 2018). In contrast, community-level bot detection should generalize to diverse communities and time periods. To this end, we propose an approach to combining existing datasets for BotPercent training. Specifically, we merge a wide variety of publicly available Twitter bot detection datasets[2], and train models on the resulting dataset. By jointly leveraging diverse and representative existing Twitter bot detection datasets as training data, BotPercent presents a community-level bot detection system designed for better generalization.

Although individual models provide scores indicating the likelihood of each account being a bot, they could not be trivially aggregated for community-level estimation since binary classifiers often produce scores that do not accurately reflect true probabilities (Platt et al., 1999; Guo et al., 2017), i.e., models are often *miscalibrated*. To accurately estimate the probability of a Twitter account to be a bot and obtain percentages of bots within communities, BotPercent performs confidence calibration for all sub-models to ensure alignment between estimated and true probabilities. Specifically, we leverage temperature scaling (Guo et al., 2017), a post-processing method that rescales confidence predictions by tuning a single scaling parameter over a held-out validation set. By repeating this calibration process for each and every modular component covering the three modalities, BotPercent results in calibrated and less biased estimators for bot population results.

## 2.3 Prediction Combination

After obtaining the calibrated results of all sub-models, BotPercent combines their predictions

---

[2]Dataset details are presented in Appendix I, datasets include CRESCI-15 (Cresci et al., 2015), GILANI-17 (Gilani et al., 2017), CRESCI-17 (Cresci et al., 2017), MIDTERM-18 (Yang et al., 2020), CRESCI-STOCK-18 (Cresci et al., 2018, 2019), CRESCI-RTBUST-19 (Mazza et al., 2019), BOTOMETER-FEEDBACK-19 (Yang et al., 2019), TWIBOT-20 (Feng et al., 2021b), and TWIBOT-22 (Feng et al., 2022b).

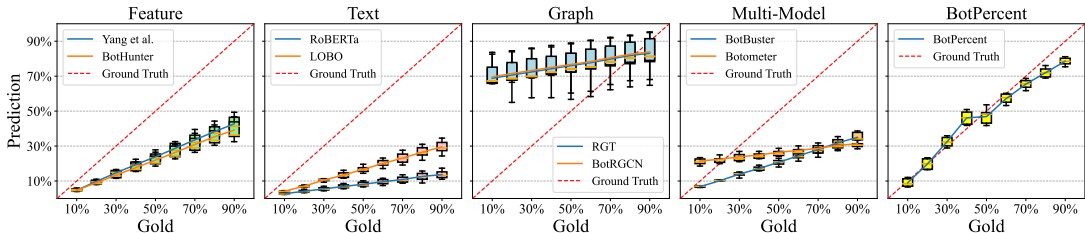

Figure 3: Predicted bot percentages on resampled communities from the TwiBot-22 benchmark that simulate imbalanced settings, with bot percentages ranging from 10% to 90%. BotPercent consistently yields accurate population estimations, while baselines severely under- or over-estimate bot populations across Twitter communities.

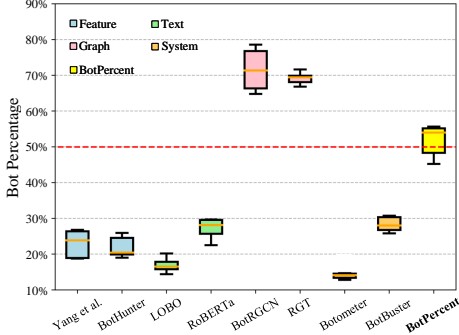

Figure 4: BotPercent and existing models' community-level bot population estimation for the 10 balanced community datasets in TwiBot-22 (ground truth is 50% for all communities). BotPercent is significantly more accurate for all communities analyzed.

through weighted summation:

$$p = \frac{1}{\mathcal{D}} \sum_{i=1}^{\mathcal{D}} \mathrm{argmax}_{\{0,1\}} \left( \sum_{j}^{\{f,t,g\}} \sum_{k=1}^{K_j} \alpha_{jk} \cdot p_{jk}^i \right)$$

where $\mathcal{D}$ denotes the number of accounts in a given Twitter community, $f, t, g$ respectively denote feature-, text-, and graph-based approaches, $K_j$ denotes the number of sub-models under the $j$-th modality, and $\alpha_{jk}$ denotes the learnable weight of the $k$-th sub-model under category $j$, which is optimized using the negative log likelihood (NLL) loss on the validation set with weights of sub-models kept frozen. The resulting $p$ is the BotPercent's bot population estimation for a community.

## 3 Experiments

### 3.1 Balanced Twitter Communities

We first evaluate BotPercent in balanced settings by leveraging the 10 Twitter communities presented in a recent TwiBot-22 benchmark (Feng et al., 2022b). Each community contains 10,000 users with an even split between genuine users and bots. As a result, the correct estimation of Twitter bot populations in these communities would be 50%. We conduct community-level bot detection with BotPercent and compare with different types of existing bot detectors: Yang et al. (2020), BotHunter (Gu et al., 2007), LOBO (Echeverría et al., 2018), RoBERTa (Liu et al., 2019b), BotRGCN (Feng et al., 2021c), RGT (Feng et al., 2022a), Botometer (Yang et al., 2022a), and Bot-Buster (Ng and Carley, 2022)[3]. Figure 4 shows that BotPercent is significantly more accurate at predicting the true bot percentage for the 10 populations analyzed, including state-of-the-art individual bot detection methods such as RGT. In addition, feature- and text-based methods generally underestimate the bot population, while graph-based methods generally overestimate the percentage of bots.

### 3.2 Imbalanced Twitter Communities

To better understand the performance of BotPercent, and ensure it does not simply learn the balanced bot distribution, we further resample imbalanced communities of 5,000 users from the TwiBot-22 (Feng et al., 2022b) benchmark based on network proximity with bot percentages ranging from 10% to 90%. We then conduct community-level bot detection on these resampled communities and compare BotPercent with representative bot detectors. Figure 3 demonstrate that BotPercent consistently outperforms baseline models by offering consistent Twitter bot population estimations that are close to the $y = x$ ground truth. These results, along with the balanced settings performance, demonstrate the importance of BotPercent as a multi-dataset multi-model pipeline to combine their inductive biases and improve generalization, as well as confidence calibration's effectiveness for accurate bot percentage estimation.

---

[3]Baseline details are presented in Appendix O.

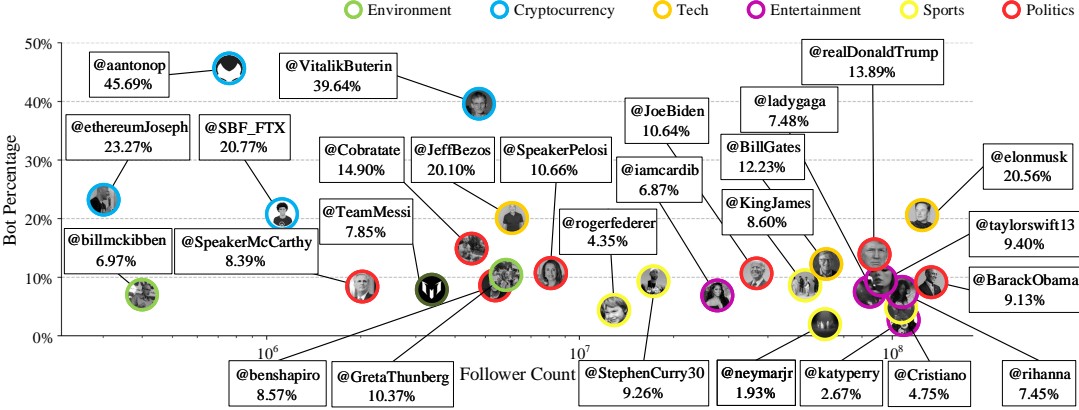

Figure 5: Bot percentage within the comment sections of celebrities from different interest domains. Celebrity-centric communities in cryptocurrency, tech, and politics are generally witnessing a larger extent of bot manipulation.

## 4  Analysis of Twitter Bots

Section 3 demonstrates that `BotPercent` achieves state-of-the-art performance on community-level bot detection and produces well-calibrated bot percentage estimation for diverse Twitter communities. We now investigate diverse real-world Twitter communities and estimate their bot populations by leveraging `BotPercent`.

### 4.1  Bot Population among User Interactions

Comment sections of famous users' tweets are battlegrounds of public opinion (Weber, 2014). As a result, we investigate the bot percentage in these comment sections and understand the extent to which celebrity-centric and news-sharing groups are compromised by Twitter bots.

### 4.1.1  Celebrities

We examine 28 celebrities from 6 interest domains: sports, entertainment, tech, politics, crypto, and environmentalism. We collect all accounts that commented on these users' tweets from December 23rd to 31st, 2022. Results show that the bot percentage in the comment sections of cryptocurrency celebrities is significantly higher than in other domains, and the bot percentage in tech is also generally above average (see Figure 5). This suggests a non-uniform spatial distribution of bots on social networks. Although previous works mainly focused on Twitter bots in the political domain (Woolley, 2016; Forelle et al., 2015), our findings reveal that Twitter bots are heavily active in various domains, particularly cryptocurrency and technology. This highlights the importance of studying bot impacts beyond politics, with implications for financial fraud, market manipulation, and more.

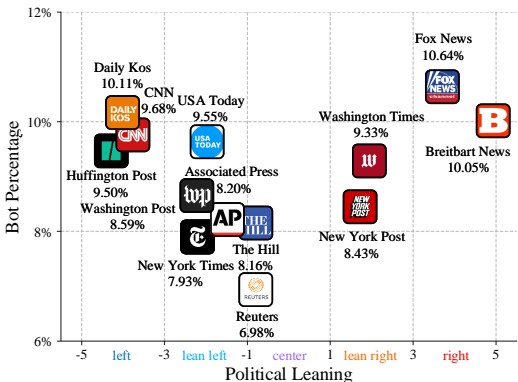

Figure 6: Bot percentage within the comment sections of news media with respect to their political leaning. Bot interference is higher in more partisan news media.

### 4.1.2  News Media

With the advent of online social networks, traditional news media have also used social media to report on current events and provide political commentary. Although previous research has shed light on bot involvement in news media (Shao et al., 2017), it remains unclear to which extent the Twitter accounts of news outlets are targeted by bots for amplification or rebuttal, which could in turn cloud the judgment of social media users. To this end, we select the official Twitter accounts of news media with different political leanings as evaluated by AllSides.[4] As shown in Figure 6, there is a strong correlation between the political stance of news media and the percentage of bot accounts in their Twitter account's comment section. Specifically, centrist news media generally have the lowest proportion of bot accounts in their comment section. As the political stance becomes more polarized,

---

[4] https://www.allsides.com

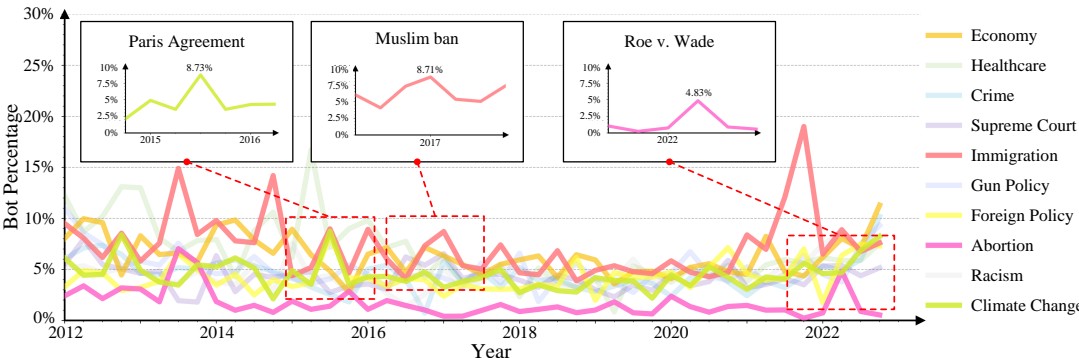

Figure 7: Temporal bot percentage trend in 11 important political issues of the past decade. Bot activity is highly correlated with major socio-political events such as Trump's Muslim ban and Roe v. Wade' overturning leaks.

the proportion of bot accounts increases, and bot percentage within the comment sections of right-leaning media is slightly higher than that of left-leaning counterparts. Based on the above analysis, we infer that news media with a centrist stance are less susceptible to interference from bot accounts in their comment section, while media with more partisan political stances are more susceptible to manipulation from bot accounts. This suggests that social media users should practice more caution when reading and interacting with hyperpartisan online news media.

## 4.2 Bot Population Changes through Time

Twitter and social media in general have become an important medium for political discourse. Worryingly, Twitter bots are often operated by malicious actors to interfere with political discussions (Caldarelli et al., 2020). To better understand the patterns of political interference from Twitter bots, we investigate 11 political topics and use political keywords presented in Flores-Saviaga et al. (2022) to search for tweets posted during different time periods and analyze the corresponding Twitter users. For each political topic, we collect tweets from 1000 users per quarter in the past decade from January 2012 to December 2022. As shown in Figure 7, the proportion of bot accounts changes in line with major socio-political events. For example, the spikes in bot participation regarding immigration in spring 2017 coincides with Trump's Muslim ban (Pierce and Meissner, 2017). Bots discussing climate change peaked in the summer of 2015, which could be attributed to the negotiations and signing of the Paris Agreement, and the sudden spike regarding abortion in early 2022 could be attributed to the Supreme Court leaks concerning

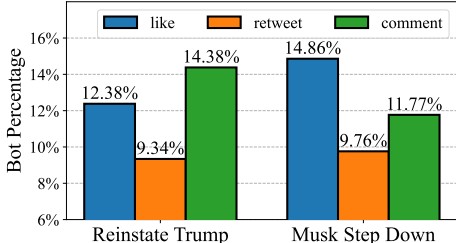

Figure 8: Bot percentage among users that interacted with the two social media moderation votes held by Elon Musk. These votes witnessed substantial interference from Twitter bots, casting doubt on Elon Musk's "Vox Populi, Vox Dei" principle.

Roe v. Wade. In addition to temporal trends, Figure 7 demonstrates that bot participation in immigration and healthcare discussions is generally higher than other topics (6.89% and 6.66%, on average). Together, these results show that Twitter bot behavior exhibits variable temporal patterns and spikes in response to major socio-political events, suggesting that content moderators and day-to-day users should practice extra caution when moderating and engaging in discussions of emerging events.

## 4.3 Bot Presence in Content Moderation Votes

Since Elon Musk's takeover of Twitter in 2022, he has held numerous votes on his personal account, with two of them having consequential content moderation outcomes: one vote to decide whether to reinstate Donald Trump[5] on Twitter and another vote to decide whether Musk should step down as Twitter CEO.[6] While the policy of direct democracy for content moderation seems straightforward,

---

[5]https://twitter.com/elonmusk/status/1593767953706921985
[6]https://twitter.com/elonmusk/status/1604617643973124097

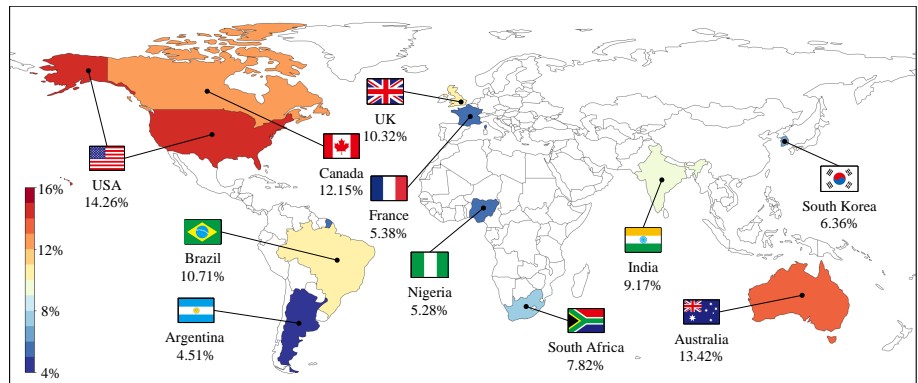

Figure 9: Bot populations within the political landscapes of different countries. The United States has the highest percentage of bots in the political community, while bot populations in English-speaking countries are generally above average.

it has numerous concerns, one of them being the interference from malicious actors through Twitter bots. We investigate the bot percentage in users that retweeted, commented, or liked the two votes (the specific voting data is not publicly available). Figure 8 shows that ∼8% to 14% of users that interacted with the two content moderation votes are bots. Given the close results of both votes (51.8% v. 48.2%, 57.5 % v. 42.5%), this suggests that bots could have changed the outcome, putting into question the validity of the "Vox Populi, Vox Dei" principle of social media moderation.

## 5 Bot Population in Different Countries' Politics

Existing research on the Twitter bot population mostly focuses on bots in U.S. politics (Bessi and Ferrara, 2016; Yang et al., 2020) while neglecting the political landscape of other countries that could have similar problems. We complement the scarce literature by investigating the bot population in different countries' political communities. Specifically, we use the president or prime minister's Twitter account as a starting point and sample their followers to serve as a proxy for the politically engaged communities in different countries. Figure 9[7] illustrates that the percentage of bots in U.S. politics is the highest, while other English-speaking countries also witness higher levels of bot interference. In addition, the percentages of bots in the political communities of Argentina, France, and Nigeria are the lowest, indicating a more genuine and authentic political discourse. These results again reaffirm that Twitter bots have spatial pat-

---

[7]Country border source: https://www.naturalearthdata.com/

| Method | Individual | | Community |
|---|---|---|---|
| | Accuracy | F1-score | Bot % |
| **BotPercent** | **0.731** | **0.726** | **+3.9%** |
| - W/O FEAT-BASED MODELS | 0.649 | 0.698 | +15.2% |
| - W/O TEXT-BASED MODELS | 0.627 | 0.701 | +9.3% |
| - W/O GRAPH-BASED MODELS | 0.659 | 0.639 | -33.8% |
| - W/O CALIBRATION | 0.653 | 0.693 | +23.8% |
| - MEAN POOLING AS COMB. | 0.708 | 0.711 | +7.05% |

Table 1: Ablation study of BotPercent. **Bold** indicates the best performance, underline indicates the second best. In community-level bot detection, we compare the model estimation to the ground truth bot percentage within 10 communities in the TwiBot-22 dataset. The ground truth is represented as 0%, with negative percentages indicating an underestimation of the bot population, and positive percentages indicating an overestimation.

terns across the whole Twitter network, while the impact of malicious Twitter bots in countries other than the U.S. warrants further research.

## 6 Ablation Study

As BotPercent outperforms various baselines in community-level bot detection tasks, we investigate the impact of each module in BotPercent to verify their effectiveness. More specifically, we perform ablation studies on feature-, text-, and graph-based modules and test the effectiveness of calibration and weighted sum combination, as is shown in Table 1. First, it is illustrated that full BotPercent outperforms 6 ablated models, proving our design choice's effectiveness. Second, we observe a significant bot percentage bias without the graph-based module version model, indicating its indispensable role in BotPercent's strong community-level bot detection performance. Finally, the community-level bot detection performance dropped dramatically without confidence calibration, indicating

that confidence calibration plays an essential role in accurate bot percentage estimation.

## 7    Related Work

Existing Twitter bot detection models mainly fall into feature-based, text-based, and graph-based.

**Feature-based** bot detection methods extract features from user timelines and metadata while using traditional classification algorithms to identify bots. Features can be extracted from various sources such as metadata (Yang et al., 2020; Lee et al., 2011), user description (Hayawi et al., 2022), tweets (Miller et al., 2014), temporal tweeting patterns (Mazza et al., 2019), and the following relationship (Feng et al., 2021c, 2022a). Later research focused on improving the scalability of feature-based approaches (Yang et al., 2020), automatically discovering new bots (Chavoshi et al., 2016), and finding the optimal balance between precision and recall (Morstatter et al., 2016). However, as bot operators become more aware of the features used by feature-based methods, they try to alter them to evade detection (Cui et al., 2020), which makes it difficult for feature-based methods to effectively identify advanced bots.

**Text-based** methods propose to leverage techniques in natural language processing to analyze tweets and user descriptions. These methods employ word embeddings (Wei and Nguyen, 2019), recurrent neural networks (Kudugunta and Ferrara, 2018), the attention mechanism (Feng et al., 2022a), and pretrained language models (Dukić et al., 2020) to analyze Twitter text. Later research efforts combine tweet representations with user features (Cai et al., 2017), learn unsupervised user representations (Feng et al., 2021a), and address the issue of multilingual tweets (Knauth, 2019). However, advanced bots counter text-based approaches by diluting malicious tweets with content stolen from genuine users (Cui et al., 2020), while Feng et al. (2021a) found that relying solely on tweet content may not be robust or accurate enough to win the bot detection arms race.

**Graph-based** methods for Twitter bot detection interpret Twitter as a network and leverage concepts from network science and geometric deep learning. These approaches adopt node centrality (Dehghan et al., 2022), representation learning (Pham et al., 2022), graph neural networks (GNNs) (Ali Alhosseini et al., 2019; Alothali et al., 2022; Yang et al., 2022b), and heterogeneous GNNs

(Feng et al., 2021c; Lei et al., 2022; Peng et al., 2022). Researchers have also explored combining graph- and text-based methods (Guo et al., 2021) or proposing new GNN architectures to take into account heterogeneities in the Twitter network (Feng et al., 2022a). These graph-based approaches have been effective in addressing the challenges such as bot clusters and bot disguises (Feng et al., 2021c).

There are also preliminary attempts to ensemble methods from different categories in bot detection. For example, Ng and Carley (2022) trains feature- and text-based experts for ensemble, and Liu et al. (2023) further ensembles three models each from feature-, text-, and graph-based categories to boost community-agnostic bot detection. However, to improve stability and generalizability, the ensemble size for both data and model needs to be increased.

Although these community-agnostic approaches have advanced the understanding of Twitter bots and their behavior, **community-level bot detection**, aiming to understand the extent to which bots compromised a given Twitter community, remains an underexplored yet important problem. With great implications for both platform moderators and everyday users, we propose a novel community-level system BotPercent with multi-dataset training and the largest model ensemble to estimate the Twitter bot populations from groups to crowds.

## 8    Conclusion

We propose BotPercent, a multi-dataset multi-model Twitter bot detection system for estimating the Twitter bot populations from groups to crowds. Experiments on the TwiBot-22 benchmark demonstrate that BotPercent achieves state-of-the-art performance on community-level bot detection while being significantly more robust to perturbations in user features. Armed with BotPercent, we investigate the bot populations in Twitter communities, such as news commentators, politically engaged accounts, and more. Together these results demonstrate that the existence of Twitter bots is not homogeneous, rather having spatial-temporal patterns that have great implications for social media moderation, warning of socio-political events, real-time monitoring, and BotPercent's potential to guide future research in specific communities rather than general-purpose.

## Limitations

We identify three key limitations. Firstly, due to the limitations of the Twitter API, our estimation of Twitter bot populations is sometimes limited in scale and information completeness. For instance, Twitter API limits the tweet lookup rate to 900 per 15 minutes, and we do not have access to user information that participated in Twitter votes. However, our work presents `BotPercent` and a series of bot analysis proposals that are compatible with better API access and improved data sources. Secondly, `BotPercent` employed knowledge distillation to make graph-based models scalable to real-world analysis, while this approach may result in minor performance drops due to the absence of graph structure information in the inference stage. Finally, `BotPercent` may introduce bias in the cold start circumstance, where users who have not engaged with the platform for a long enough time are more likely to be identified as bot accounts.

## Ethics Statement

We envision `BotPercent` as a large-scale pre-screening tool and not as an ultimate decision maker. Importantly, `BotPercent` and any other automatic bot detection systems are imperfect proxies for bot detection and thus need to be used with care, in collaboration with human moderators to monitor or suspend suspicious accounts. Apart from that, as a combination of datasets and models, `BotPercent` may inherit the biases of its constituents. For example, pretrained language models could encode undesirable social biases and stereotypes (Nadeem et al., 2021; Liang et al., 2021; Bender et al., 2021), while graph neural networks could also discriminate against certain demographic groups (Dong et al., 2022) in decision-making. We leave to future work on how to incorporate the bias detection and mitigation techniques developed in ML research in bot detection systems.

## Acknowledgements

We thank the reviewers and area chair for their comments and feedback. This research is supported in part by the Office of the Director of National Intelligence (ODNI), Intelligence Advanced Research Projects Activity (IARPA), via the HIATUS Program contract #2022-22072200004. This material is funded by the DARPA Grant under Contract No. HR001120C0124. We also gratefully acknowledge support from NSF CAREER Grant No. IIS2142739, and the Alfred P. Sloan Foundation Fellowship. The views and conclusions contained herein are those of the authors and should not be interpreted as necessarily representing the official policies, either expressed or implied, of ODNI, IARPA, or the U.S. Government. The U.S. Government is authorized to reproduce and distribute reprints for governmental purposes notwithstanding any copyright annotation therein.

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

## A  Discussion

In this work, we propose the novel setting of community-level bot detection, design the `BotPercent` pipeline, and investigate the bot population from groups to crowds. Our experiment and results demonstrate that the distribution of Twitter bots is not homogeneous, but rather possesses spatial-temporal patterns. The *spatial* and *temporal* nature of the bot populations has great implications for social media moderation, day-to-day users, and future research.

Since the Twitter bot population has *spatial* patterns, it could be beneficial to invest more resources into high-stakes communities with higher bot presence. For example, in this work, we demonstrated that the bot percentage in political and cryptocurrency communities is much higher than in entertainment. This finding suggests that in addition to building general-purpose bot detection systems, the fine-grained study of these "highly polluted" communities (Uyheng and Carley, 2020) is also of integral importance.

Since the Twitter bot population has *temporal* patterns, it could be beneficial to take socio-political events into account when investing content moderation resources. For example, in this work, we showed that bot presence in the online discussion regarding immigration and healthcare often spikes around major U.S. elections. This finding suggests that informing social media users of potential interference and curbing the malicious impact of Twitter bots is especially important during elections (Bessi and Ferrara, 2016), referendums (Stella et al., 2018), and other major socio-political events (Woolley, 2016).

## B  Problem Definition

Let $u = (u_f, u_c)$ be a social media account, where $u_f$ represents all its accessible features (metadata, textual, graph-based) and $u_c \in \{\text{human}, \text{bot}\}$ represents whether the account is managed by a human or a bot. Let $\mathcal{U}$ be a group of users, and let $p_{\mathcal{U}} = |\{u : u \in \mathcal{U}, u_c = \text{bot}\}| \, / \, |\mathcal{U}|$ the true proportion of bots in the population $\mathcal{U}$. Community-level bot detection targets to learn a precise estimator $\widehat{p_{\mathcal{U}}}$ of $p_{\mathcal{U}}$, *i.e.* aiming to minimize $|p_{\mathcal{U}} - \widehat{p_{\mathcal{U}}}|$.

## C  Feature Perturbation Study

While existing Twitter bot detection approaches heavily rely on the verified (*i.e.* blue check mark) as

| Model | Individual (F1-score) | | | Community (bot %) | | |
|---|---|---|---|---|---|---|
| | All True | All False | Random | All True | All False | Random |
| Yang et al. | 0.518 | 0.640 | 0.583 | 17.80% | 23.96% | 20.89% |
| LOBO | 0.272 | 0.476 | 0.380 | 8.01% | 16.20% | 12.08% |
| BotRGCN | 0.003 | 0.640 | 0.466 | 0.01% | 84.07% | 42.03% |
| RGT | 0.002 | 0.639 | 0.464 | 0.04% | 83.23% | 41.58% |
| BotBuster | 0.000 | 0.558 | 0.341 | 0.00% | 23.21% | 11.52% |
| **BotPercent** | **0.656** | **0.672** | **0.665** | **47.67%** | **59.94%** | **55.21%** |

Table 2: Feature perturbation study where we alter user verification to all True, all False, or randomly assigned. We report F1-score for individual analysis and the bot percentage for community-level detection, where the ground truth is 50% and the closer the better.

| Hyperparameter | Graph-Based | Text-Based | GNN Distillation |
|---|---|---|---|
| LEARNING RATE | $1 \times 10^{-3}$ | $1 \times 10^{-4}$ | $5 \times 10^{-4}$ |
| BATCH SIZE | 128 | 64 | 2048 |
| EPOCHS | 50 | 50 | 50 |
| L2 REGULARIZATION | $1 \times 10^{-5}$ | $1 \times 10^{-5}$ | $1 \times 10^{-5}$ |
| HIDDEN DIM | 128 | 128 | 1024 |
| DROPOUT | 0.5 | 0.5 | 0.3 |
| LAYER COUNT $L$ | 2 | - | 2 |
| $\lambda$ | 0.7 | - | - |

Table 3: Hyperparameter settings of `BotPercent`.

an important indicator (Yang et al., 2020), recently there were significant changes to Twitter's verification policy: existing verified users might lose their verified status, while previously unverified users could get a blue checkmark by subscribing to Twitter Blue. This has great implications for Twitter bot detection since verification was a widely adopted and essential feature across multiple types of bot detectors. As a result, a desirable bot detection system should be robust to such feature perturbations, especially for the verified binary feature. To this end, we evaluate `BotPercent` and baselines on three new settings: a) all users become verified users, b) all users become unverified users, and c) the user verification status is randomly assigned. This is to imitate a scenario where user verification is no longer reliable and how bot detectors would fare in this case. We present the results in Table 2, which demonstrates that disabling the verification feature would seriously cripple the performance of existing bot detection systems. On the contrary, `BotPercent` maintains steady performance both in individual (determing the bot-or-not of individual users) and community-level approaches, thanks to its multi-modal and multi-model pipeline reducing the over-reliance on the verified feature.

## D  Bot Percentage among Active Users

We also provide an answer to an important and widely debated question: the overall percentage of bots among *active* Twitter users. We define ac-

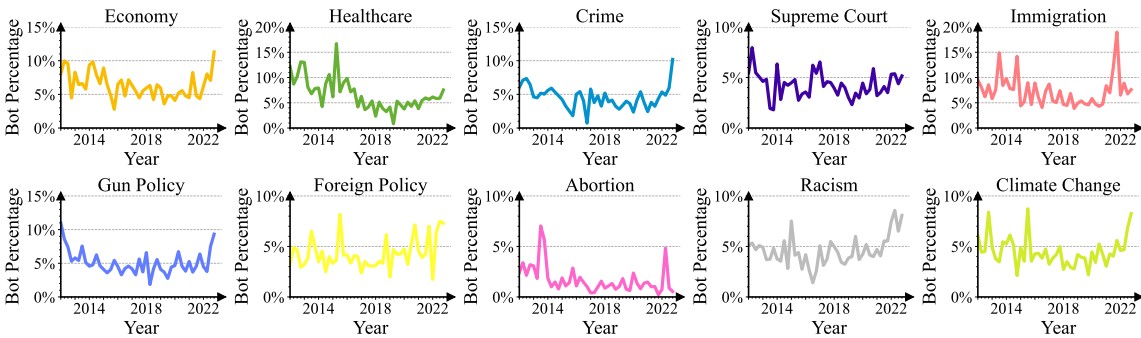

Figure 10: The temporal trends of the bot percentage on the 11 political issues over the past decade.

| Model | Accuracy | F1-score |
|---|---|---|
| Yang et al. (Yang et al., 2020) | 0.623 | 0.395 |
| BotHunter (Gu et al., 2007) | 0.614 | 0.370 |
| LOBO (Echeverria et al., 2018) | 0.552 | 0.198 |
| RoBERTa (Liu et al., 2019a) | 0.633 | 0.432 |
| BotRGCN (Feng et al., 2021c) | 0.488 | 0.485 |
| RGT (Feng et al., 2022a) | 0.509 | 0.509 |
| Botometer (Yang et al., 2022a) | **0.755** | 0.585 |
| BotBuster (Ng and Carley, 2022) | 0.627 | 0.439 |
| **BotPercent** (Ours) | 0.731 | **0.726** |

Table 4: Individual bot detection performance. BotPercent achieves a significantly higher F1-score thanks to its balanced and well-calibrated predictions.

tive users as those who generated content on the platform (posts, comments, retweets) and exclude passive users (those that solely browsed or liked posts). This may be seen as a proxy to understand how much of the genuine users' content consumption comes from bot accounts.

## E Individual Bot Detection

In addition to achieving state-of-the-art performance on community-level bot detection, we also evaluate BotPercent at the individual level. We leverage the expert annotated dataset in the TwiBot-22 benchmark. BotPercent outperforms all baselines by at least 14.1 F1-score (see Table 4). While Botometer achieves slightly higher accuracy, further examination shows that it greatly underestimates the bot population (also shown in Figure 4) by skewing its predictions heavily towards genuine users, which is also evident in its lower F1-score results. In summary, BotPercent achieves more balanced predictions and yields an overall improved performance.

## F Details of Bot Percentage Trend in Different Topics

To facilitate readability, we divide Figure 7 into 10 topics and individually present trends in Figure 10.

## G Communities Details

We adopt the 10 community datasets proposed in the TwiBot-22 dataset (Feng et al., 2022b) to evaluate BotPercent in Section 3.1, which start from five closely connected sub-communities around *@BarackObama*, *@elonmusk*, *@CNN*, *@NeurIP-SConf*, and *@ladygaga*. Then, Feng et al. (2022b) used K-means to cluster the Word2Vec (Mikolov et al., 2013) representations of hashtags and identified users tweeting about similar hashtags into five additional sub-communities.

## H Hyperparameter Details

The hyperparameters of BotPercent are presented in Table 3 to facilitate reproducibility.

## I Dataset Details

The CRESCI-15 (Cresci et al., 2015) dataset mainly consists of accounts collected from a volunteer base and active Italian Twitter users. Users in the GILANI-17 (Gilani et al., 2017) dataset are collected with the Twitter streaming API and are grouped into four categories based on the number of followers. CRESCI-17 features three types of bots: traditional spambots, social spambots, and fake followers. The MIDTERM-18 (Yang et al., 2020) dataset is filtered based on political tweets and active users collected during the 2018 U.S. midterm elections. For the CRESCI-STOCK-18 (Cresci et al., 2018, 2019) dataset, bot users were identified by finding accounts with similar timelines among tweets containing the selected hashtags during five months in 2017. The

| User Metadata | Derived Feature | Calculation |
|---|---|---|
| STATUS_COUNT | SCREEN NAME/USERNAME DIGITS | No. digits in screen name or username digits |
| FOLLOWER_COUNT | TWEET_FREQUENCY | status_count / user_age |
| FRIEND_COUNT | URL_COUNT | No. URL in user descriptions |
| FAVORITE_COUNT | BOT WORD IN DESCRIPTION/SCREEN NAME/USERNAME | No. "bot" in description/screen name/username string |
| LISTED_COUNT | USERNAME ENTROPY | $-\sum_{i=1}^{n} p_i \log_2 p_i$, where $p_i$ is the normalized string count |
| DEFAULT_PROFILE | SCREEN NAME/USERNAME/DESCRIPTION_LENGTH | length of name string |
| PROFILE_USE_BACKGROUND_IMAGE | FOLLOWERS_GROWTH_RATE | followers_count / user_age |
| VERIFIED | FRIENDS_GROWTH_RATE | friends_count / user_age |
| USER_ID | SCREEN NAME/DESCRIPTION_HASHTAG_COUNT | No. hashtag in string |
| PROTECTED | FOLLOWER_FRIEND_RATIO | follower_count / friends_count |
| HAS_LOCATION | USERNAME_CAPITAL_LETTER_COUNT | No. capital letter in username |
| USER_AGE | SCREEN NAME/USERNAME UNICODE GROUP | group user's screen name and username to 105 Unicode groups |
| | DESCRIPTION_SENTIMENT_SCORE | sentiment scores generated by VADER (Hutto and Gilbert, 2014) |
| | USERNAME & SCREEN NAME DISTANCE | Levenshtein distance (Levenshtein et al., 1966) between two strings |

Table 5: Features adopted in the feature-based module in BotPercent. User metadata features are directly extracted from Twitter API, and derived features are calculated based on user metadata.

| category | sub-model | weight $\alpha$ | sum |
|---|---|---|---|
| **feature-based** | Random Forest | 0.544 | 1.127 |
| | Adaboost | 0.583 | |
| **text-based** | RoBERTa | 0.404 | 0.815 |
| | T5 | 0.411 | |
| **graph-based** | HGT | 0.247 | 0.852 |
| | SimpleHGN | 0.205 | |
| | BotRGCN | 0.192 | |
| | RGT | 0.208 | |

Table 6: Learned weight $\alpha_{jk}$ for each sub-model (Section 2.3), where $j \in \{f, t, g\}$ denotes feature-, text-, and graph-based methods respectively.

CRESCI-RTBUST-19 (Mazza et al., 2019) dataset was crawled from Italian retweets between 17-30 June 2018. The BOTOMETER-FEEDBACK-19 (Yang et al., 2019) dataset was constructed by manually labeling accounts annotated by feedback from Botometer users. TWIBOT-20 (Feng et al., 2021b) consists of users from four interest domains collected from July to September 2022. TWIBOT-22 (Feng et al., 2022b) uses diversity-aware BFS to collect users by expanding with the follow relationships. To prevent test data leakage, we removed user labels for the 10 communities, as well as the expert-labeled users in the TwiBot-22 dataset. Moreover, we created a dictionary structure projecting from user ID to their account information to prevent test data leakage caused by user overlap. This ensures no overlap between the training data in the merged version and the data used for evaluation.

## J Feature Details

To facilitate further research, we present the list of features used in the feature-based module of BotPercent in Table 5.

## K Learned Weights for Each Model

To gain a better understanding of the importance of each module and its implications in BotPercent, we present the learned weights for each sub-model in Table 6. We found that feature-based methods received the highest weight, while text-based methods received the lowest weight. This indicates that feature-based methods have the greatest impact on the prediction results, while graph-based methods have a slightly greater impact than text-based methods.

## L Computation Details

We used a server with 8 NVIDIA 2080 Ti GPUs, 178GB memory, and 24 CPU cores for BotPercent training. The inference stage is conducted on a PC with 4 CPU cores and 16GB memory. Training BotPercent with the best hyperparameters takes approximately 1 hour on the merged dataset.

## M Scientific Artifacts

BotPercent is built with the help of many existing scientific artifacts, including PyTorch (Paszke et al., 2019), torch geometric (Fey and Lenssen, 2019), Tweepy (Roesslein, 2009), Numpy (Harris et al., 2020), sklearn (Pedregosa et al., 2011), and transformers (Wolf et al., 2020). We have made the BotPercent implementation publicly available.

## N Active-User Collection Methodology

We use the StreamClient function in the Twitter API to sample 1% of real-time tweets and corresponding users from December 12th to 18th, 2022. The StreamClient Twitter API function only provides posts or retweets but does not include users who only liked or browsed Twitter content.

## O   Baseline Details

- **Yang et al. (2020)** leverages 8 types of user metadata and 12 derived features to identify bot accounts with random forest.

- **BotHunter** (Gu et al., 2007) leverages random forest to classify users based on account metadata, network attributes, content, and timing information.

- **LOBO** (Echeverrıa et al., 2018) extracts 26 user features and feeds them into random forest for classification.

- **RoBERTa** (Liu et al., 2019a; Feng et al., 2022b) leverages RoBERTa to encode user tweets and descriptions and feed them into an MLP to distinguish bots from humans.

- **BotRGCN** (Feng et al., 2021c) utilizes text information from user descriptions and tweets, and numerical and categorical user property information as user features. Then BotRGCN constructs a heterogeneous graph from the Twitter network based on user relationships and applies relational graph convolutional networks (RGCN) for bot classification.

- **RGT** (Feng et al., 2022a) uses graph transformers and semantic attention network to model the intrinsic influence heterogeneity and relation heterogeneity in Twittersphere. RGT uses the same input as BotRGCN and feeds user features into RGT layers for bot identification.

- **Botometer** (Yang et al., 2022a) is a public website to check the activity of a Twitter account and returns a score, where higher scores mean more bot-like activity. Botometer system leverages more than 1,000 features using available metadata and information extracted from interaction patterns and content.

- **BotBuster** (Ng and Carley, 2022) enhances cross-platform bot detection by processing user metadata and textual information with the mixture-of-experts architecture.