# OpenReview forum: "BotPercent: Estimating Bot Populations in Twitter Communities"
_EMNLP/2023/Conference — EMNLP 2023 Findings_

### Official Review · Reviewer_Rurt · 2023-08-04

**Soundness:** 3

**Excitement:**

4: Strong: This paper deepens the understanding of some phenomenon or lowers the barriers to an existing research direction.

**Paper Topic And Main Contributions:**

This paper proposes a multi-dataset multi-model approach based on user metadata and derived features, written text, and network structure for estimating the percentage of bots in Twitter communities. The authors combine and calibrate the modular components and compare the performance of their proposed bot detection with a recent benchmark (TwiBot-22) and achieve state-of-the-art performance. They also investigate the ratio of bots in several Twitter communities, such as those who have commented on celebrities' tweets, news media, or content moderation polls posted by Elon Musk on Twitter.

**Questions For The Authors:**

Q-A: In section 4.3, did you check for overlaps among these accounts? Overlaps among the accounts that have had any of the three types of engagements and overlaps among the accounts that have interacted with either of these two polls.
Q-B: In section 4.3, the authors have concluded that these percentages of bots could have changed the outcomes of the two polls. This means that most of the 8–14% of accounts identified as bots have voted for the reinstatement of Donald Trump on Twitter (since this is the only way the outcome could have changed). On the other hand, if 8–14% of accounts that have voted for Elon Musk Step Down are bots, most of them should have been against Elon Musk being the CEO (this is the only way they could have changed the outcome. Since currently, the poll asks him to step down and the supporters of stepping down are the winners of the poll). But, if Elon Musk as the CEO paves the way for Trump to reinstate, why should they change the outcome in the next poll (one month after the first) to step down? :)
Assuming the analysis stated in the previous question (Q-A) is not done (at least, it is not reported in the present paper), I think such a claim cannot be realistic.
Q-C: In section 4.3, what is the distribution (number, not percentages) of users in each poll?

**Reasons To Accept:**

A1: The presented work achieves state-of-the-art performance and performs fine on detecting bot percentages in the evaluated communities and datasets
A2: The paper uses a variety of features, models, and tools from the literature to build a reliable bot detector and compares its performance with a benchmark
A3: The paper is well-written, and the figures and tables have proper qualities
A4: The authors have investigated several different communities in different domains to evaluate their approach and provide insights

**Reasons To Reject:**

R1: The paper is too long, and important discussions such as the datasets, names and descriptions of features are left for the Appendix.
R2: There are many repetitions and verbose language in the presented paper. For example, concepts like community-based bot detection and calibration are repeated several times before actually getting to the points the paper explains its contribution towards each.
R3: While the aggregate/overall bot percentages for 10 populations/communities are presented in the paper, it is not clear how the BotPercent's individual bot detection performs in each of these communities and the whole datasets. For example, if the bot percentage is 40% instead of 50%, what portion of those 40% of accounts identified as bots are really bots? Such analysis would clarify to what extent the achieved performance on the overall bot percentage is based on correct detection of bots (true positives), and to what extent is because of false positives.

**Reproducibility:**

4: Could mostly reproduce the results, but there may be some variation because of sample variance or minor variations in their interpretation of the protocol or method.

**Reviewer Confidence:**

4: Quite sure. I tried to check the important points carefully. It's unlikely, though conceivable, that I missed something that should affect my ratings.

**Typos Grammar Style And Presentation Improvements:**

T1: In line 223, I think "i-th layer" should be "l-th layer"
T2: Figure 4 (section 3.1) comes before Figure 3 (section 3.2) in the text
T3: Figure 1 is never referred to
T4: Line 485, "Figure 8 shows that ∼8% to 14% of users that interacted with the two content moderation votes are bots." Based on the figure, is it not 9% to 15%? If not (for example, if you take weighted averages on the three types of interactions), please clarify the percentages.

---

> ### Author Rebuttal · Authors · 2023-08-28
>
> Thank you for your constructive feedback!
>
> > Presentation problem: too long, repetition, and verbose language
>
> Thanks for your suggestion! We will streamline the introduction, methodology description, as well as experiments in the final version.
>
> > The individual-level bot detection performance is not clear. To what extent the performance is based on the correct prediction, like true positive and false positive?
>
> In Appendix F, we present the individual-level bot detection performance (Acc and F1) on the expert annotated dataset in TwiBot-22. In the final version, we will include the individual-level bot detection performance metrics, such as recall and precision to assess the extent to which the achieved performance on the overall bot percentage is based on the correct detection of bots.
>
> > In section 4.3, did you check the overlaps between either of the two interaction groups?
>
> We conducted the experiment, and the results are presented in the table below (in user count):
>
> |    | like & retweet  | like & comment | comment & retweet|
> |  ----  | :----:  | :----:  | :----:  |
> | **Reinstate Trump vote**  | 801 | 20 | 20 |
> | **Musk step-down vote**  | 892 | 48 | 32 |
>
> > In section 4.3, if the identified percentage of bot accounts could have changed the outcomes of the two polls, it would imply that most of the bots voted for the reinstatement of Donald Trump and against Elon Musk as CEO, which seems contradictory and unrealistic.
>
> We examined the overlap between users who participated in the Reinstate Trump vote and the Musk step-down vote. Out of the 37,498 and 38,506 users involved, only 536 users overlapped, accounting for 0.91% and 0.88% respectively. This statistic suggests that the users or bots participating in both votes differ significantly, potentially holding contrasting opinions and voting for different stances. Consequently, we believe our claim that "bots could have influenced the outcome" to be reasonable.
>
> To provide further context, as of 2023, Twitter has 353.9 million active users [1]. Based on our findings, approximately 28 million bot accounts exist, while the users engaged in these votes amount to 15 million and 17 million respectively. This further serves as evidence that the users involved in these two votes can be distinct, with minimal overlap.
>
> [1] https://www.statista.com/statistics/303681/twitter-users-worldwide/
>
> > In section 4.3, what’s the distribution of users in each pool?
>
> The user count distribution is:
>
> |    | retweet  | comment | like |
> |  ----  | :----:  | :----:  | :----:  |
> | **Reinstate Trump vote**  | 11,079 | 5,634 | 21,626 |
> | **Musk step-down vote**  | 12,172 | 8,519 | 18,773 |
>
> > Typo in line 223, line 485, and figure order
>
>  Thank you for pointing out! We have fixed these typos.

---

### Official Review · Reviewer_q2rU · 2023-08-04

**Soundness:** 4

**Excitement:**

4: Strong: This paper deepens the understanding of some phenomenon or lowers the barriers to an existing research direction.

**Paper Topic And Main Contributions:**

This paper studies the problem of bot population detection on Twitter, which aims to determine the proportion of bots as compared to the tradition task of detecting whether a single user is a bot. To address this problem, the paper proposed BotPercent a multi-modality neural network that combines different architectures for different input types, e.g., random forests and AdaBoost for user features, RoBERTa and T5 for txt-based features, and various GNNs for the user network/graph. There is a comprehensive set of experiments showing the good performance of BotPercent against various baselines, as well as various interesting studies into the bot population on Twitter over time and across different interest groups.

I wish the authors all the best as they proceed with this work.

Post-rebuttal: I thank the authors for their responses and acknowledge that they have been read. My scores remain unchanged.

**Questions For The Authors:**

Please refer to W1 and W2 in my detailed comments above.

**Reasons To Accept:**

There are a few strong points about this paper that I particularly like, which are:

S1.	This paper studies the important problem of bot detection on Twitter, or more specifically the proportion of bots given a specific community. This is an interesting aspect of the bot detection problem but also one that needs more motivation. For instance, detecting individual users as bots allow for actions such as banning or suspending that account but know the proportion of bots allows for less direct actions, although is still useful in some ways.

S2.	The proposed BotPercent model makes good use of multiple modality or input types of user features, text features and graph networks. Within each component, there is also a fair amalgamation of techniques for that specific domain, e.g., various types of GNN.

S3.	There is a fairly comprehensive set of experiments on a relatively large bot dataset, with various bot configurations being studied such as a bot distribution from 10% to 90%. In these experiments BotPercent shows good performance over various baselines including the ones that uses a single input type or modality.

S4.	In addition to the comparison experiments, there are also various interesting investigations into the distribution of Twitter bot population over different time periods and across different interest groups.

S5.	In general, this is a nicely written and well structured paper that is overall a pleasure to read.

**Reasons To Reject:**

There are a few weak aspects of the paper that the authors can clarify or improve on, which are:

W1.	There is a good consideration of experiments based on two settings, a balanced 50:50 bot:real user setting and various bot percentages of 10% to 90%. Given that the proposed model includes a GNN component that uses the social links, how do you ensure that any sampling of bots for this percentage do not (either favourable or unfavourably) affect the proposed model or any baselines that uses graph information?

W2.	While the BotPercent model is well-described, certain architecture design considerations or intuitions are missing from the paper. For example, how are the 12 direct and 14 derived user features identified and selected, and what effect will another user feature set have? Similarly, how would another language model (e.g., DistilBERT instead of RoBERTa) or a specific GNN or GNN combination work for the text-based and graph-based ocmponents, respectively? An ablation study might be useful for investigating some of these factors.

**Reproducibility:**

5: Could easily reproduce the results.

**Reviewer Confidence:**

4: Quite sure. I tried to check the important points carefully. It's unlikely, though conceivable, that I missed something that should affect my ratings.

---

> ### Author Rebuttal · Authors · 2023-08-28
>
> Thank you for your constructive feedback!
>
> > How to ensure the sampling does not affect the graph information?
>
> As discussed in line 352 to 358, we conducted downsampling on the 50:50 dataset and got user communities with bot percentages ranging from 10% to 90%.
>
> For our proposed BotPercent, thanks to our design choice that distills GNNs into MLP, we could conduct efficient and scalable inference without social links. That being said, BotPercent doesn’t need graph structure (social links) and is immune to the change of graph structure in the inference stage.
>
> Regarding the graph-based baseline models, the underlying graph structure remains unchanged throughout the experiments. The selection of users within a specific community for analysis does not alter the graph structure but rather serves as a means to evaluate the model's predictions within that particular context.
>
> > Ablations on user features and GNN/LM choice are needed
>
> Thanks for pointing this out. We did an ablation study on feature-, text-, graph-based models, confidence calibration, and weighted sum aggregation in Table 1, Appendix. We agree that it would be beneficial to extend this ablation study to include the analysis of user features, different language models (e.g., DistilBERT), and specific GNN architectures or combinations for the text-based and graph-based components, respectively. We will add these ablation settings in the final version.

---

### Official Review · Reviewer_TcPZ · 2023-08-05

**Soundness:** 3

**Excitement:**

2: Mediocre: This paper makes marginal contributions (vs non-contemporaneous work), so I would rather not see it in the conference.

**Missing References:**

1. I think literature on lock-step behavior from data mining research could be interesting here.


**Paper Topic And Main Contributions:**

The paper is centered on estimating the number of bots (percentage) within a certain community. The authors claim that the current bot detection methods are unimodal and more importantly, do not take into account communities or social environment into consideration while trying to find bots. The paper then goes on to use multiple different unimodal models (feature , text, graph-based) and calibrate their output scores. This proposed framework is used to estimate percentage of bots in different communities.

**Questions For The Authors:**

A. Can you elaborate on the utility of "community" or running bot-detection method on a certain "community" ? Is it rendering some information that is helpful to predict if a particular account is Bot or not. If yes, will this method fail for uncohesive or arbitrary community ?

B. The jointly working group attacks are often well studied in data-mining literature, which means multiple accounts working together in a lock-step behavior. Does it make sense to compare and contrast with that literature as well ?

C. [Individual vs Aggregate Metric] I would say though the aggregate metric i.e. percentage of bots in a community is interesting, but it is more investigative than intervention or action-oriented. Though the method performs well on both individual and aggregate metric, but I would want to know the motivation behind choosing a more investigative metric rather than more action-oriented metric to optimize on. I treat this not as a weakness especially the method works well on both of the said metrics. But if it wouldn't have then I would have preferred to opt in on more intervention based rather than investigative metric.

**Reasons To Accept:**

1. The paper is well-written and easy to follow.
2. The proposed method is an ensemble on top of existing approaches and then confidence is calibrated to identify if a particular account is a bot or not. The method outperforms multiple other baselines.
3. The analysis section is very insightful and could have wide-spread impact.

**Reasons To Reject:**

The paper is centered around an important topic, but there are some drawbacks in the paper.

[Training Dataset] The training dataset is an amalgamation of all merged datasets. One of the valid things to test out is that if there is any data leakage (temporal or other) between the evaluation set and the training set.

[Definition of Community] The definition of community is somehow unclear in the experiments. The authors in most of the settings assume that the communities are given - or are estimated through the comment section of influential accounts. It is harder to understand if having an approach i.e. "community-specific" adding any value. Let's say if I arbitrarily group a set of users as "community" - will the estimates within that community be worse than any other actual "community" ?

[Performance Gains] It is unclear if the performance gains are coming from ensemble of the models or some "community-level" calibration. Ideally the ensemble itself is a good contribution - but doesn't warrant enough to be a scientific article where the goal is to propose or unlock a new understanding. It would be ideal to see performance comparison among different ensembles such as BotRGCN or BotBuster or something else etc. It might also be helpful to clarify if how similar are individual methods in BotPercent to other baselines.

**Reproducibility:**

4: Could mostly reproduce the results, but there may be some variation because of sample variance or minor variations in their interpretation of the protocol or method.

**Reviewer Confidence:**

4: Quite sure. I tried to check the important points carefully. It's unlikely, though conceivable, that I missed something that should affect my ratings.

---

> ### Author Rebuttal · Authors · 2023-08-28
>
> Thank you for your constructive feedback!
>
> > Data leakage between evaluation and training set
>
> To prevent test data leakage, we removed user labels for the 10 communities, as well as the expert labeled users in the TwiBot-22 dataset. Moreover, we created a dictionary structure projecting from user ID to their account information to prevent test data leakage caused by user overlap. This ensures no overlap between the training data in the merged version and the data used for evaluation.
>
> > The definition of community is not clear
>
> Following previous works [1-2], we define Twitter communities by network proximity. Specifically, we evaluate BotPercent on the 10 Twitter communities collected in [1], which were obtained using BFS search and cluster techniques. The TwiBot-22 work examined that these proximity-based communities do form loosely connected interest groups, thus we follow this definition. We will add this clarification to Section 3.
>
> Regarding the utility of "community", our design choice of distilling the graph neural network into an MLP allows BotPercent to effectively fit into arbitrary user groups or communities and estimate the bot population within them. This means that the method can be applied to different user groups, regardless of how they are defined.
>
> In terms of data, the 10 user communities were obtained in the TwiBot-22 paper using BFS search and cluster techniques. For a more comprehensive understanding of the process, Please refer to Appendix H for detailed information.
>
> [1] Feng, S., Tan, Z., Wan, H., Wang, N., Chen, Z., Zhang, B., ... & Luo, M. (2022). TwiBot-22: Towards graph-based Twitter bot detection. Advances in Neural Information Processing Systems, 35, 35254-35269.
> [2] Feng, S., Wan, H., Wang, N., Li, J., & Luo, M. (2021, October). Twibot-20: A comprehensive twitter bot detection benchmark. In Proceedings of the 30th ACM International Conference on Information & Knowledge Management (pp. 4485-4494).
>
> > Performance gain from ensemble and calibration is not clear
>
> In Appendix C, we present an ablation study to investigate the factors contributing to the performance gains of BotPercent. The study involves removing feature-, text-, graph-based methods, and confidence calibration individually to assess their impact. Our findings reveal that the uncalibrated ensemble performs 19.9% worse than the calibrated ensemble. Moreover, the calibrated ensemble outperforms the calibrated model without feature-, text-, and graph-based models by 11.3%, 5.4%, and 29.9%, respectively. Results indicate that both ensemble and confidence calibration contribute significantly to the performance gain on community-level Twitter bot detection.
>
> > Comparison with the joint working group attacks and lock-step behavior literature
>
> Thanks for this great suggestion! We will discuss the joint working group attack and lock-step behavior literature, e.g.  [1, 2], in the final version.
>
> [1] Jiang, M., Cui, P., Beutel, A., Faloutsos, C., & Yang, S. (2016). Inferring lockstep behavior from connectivity pattern in large graphs. Knowledge and Information Systems, 48, 399-428.
>
> [2] Beutel, A., Xu, W., Guruswami, V., Palow, C., & Faloutsos, C. (2013, May). Copycatch: stopping group attacks by spotting lockstep behavior in social networks. In Proceedings of the 22nd international conference on World Wide Web (pp. 119-130).
>
> > Individual or aggregate metric
>
> The choice of an aggregate metric, such as the percentage of bots in a community, stems from the need to gain a comprehensive understanding in situations where data access is limited and the dataset size is immense. In these cases, an aggregate metric allows us to probe and analyze the global perspective more effectively.

---

### Meta-Review · Area_Chair_RLzm · 2023-09-15

**Recommendation:** 4

**Metareview:**

The paper is centered on estimating the number of bots (percentage) within a certain community. The proposed method is an ensemble on top of existing approaches and then confidence is calibrated to identify if a particular account is a bot or not.

The proposed approach is interesting although it shows some limitations and drawbacks:
1) It is not clear how the community has been defined in the evaluation set.
2) Related work should be extended by considering more recent works on data mining field.
3) The authors need to explain how they ensure any sampling of bots for both experimental analysis.

---

### Decision · Program_Chairs · 2023-10-07

**Decision:**

Accept-Findings

**Comment:**

The paper is centered on estimating the number of bots (percentage) within a certain community. The proposed method is an ensemble on top of existing approaches and then confidence is calibrated to identify if a particular account is a bot or not.

The proposed approach is interesting although it shows some limitations and drawbacks:
1) It is not clear how the community has been defined in the evaluation set.
2) Related work should be extended by considering more recent works on data mining field.
3) The authors need to explain how they ensure any sampling of bots for both experimental analysis.